# Study on the Design Strategy of Rehabilitation Space for Patients with Cognitive Disorders Based on the Environmental Adaptation of Disease Symptoms

**Weicong Li** [1] , **Zhaoming Du** [2,*], **Doris Hooi Chyee Toe** [1,3,*] , **Yueling Liu** [4], **Kum Weng Yong** [3,5] **and Haopai Lin** [6]

1   Faculty of Built Environment and Surveying, Universiti Teknologi Malaysia, Johor Bahru 81310, Johor, Malaysia
2   College of Arts, South China Agricultural University, Guangzhou 510642, China
3   Disaster Preparedness and Prevention Centre, Malaysia-Japan International Institute of Technology, Universiti Teknologi Malaysia, Kuala Lumpur 54100, Malaysia
4   College of Creativity and Design, Guangzhou Huashang College, Guangzhou 511300, China
5   KW Yong Architect, Seremban 70300, Negeri Sembilan, Malaysia
6   School of Art and Design, Guangdong University of Finance and Economics, Guangzhou 510320, China
*   Correspondence: dzm999@163.com (Z.D.); doristhchyee@utm.my (D.H.C.T.)

**Abstract:** Under the dual pressure of the large number of patients and the funding of expensive treatments, the established medical model is no longer able to meet the treatment needs of patients with cognitive disorders. Cognitive disorders cannot be cured, and the proposed MCI stage provides a window of opportunity for early intervention of the condition. Mild cognitive impairment (MCI) is a high-risk potential conversion state prior to a diagnosis of cognitive disorder, where the person still has the ability to live but with the presence of cognitive damage. The theory of environment-facilitated rehabilitation has begun to be applied to the study of cognitive disorders prevention, but its effectiveness and the drivers of its pathological characteristics remain unclear. In this paper, we explore spatial design strategies for the rehabilitation of patients with cognitive disorders based on the adaptability of pathological characteristics, and provide new ideas for spatial interventions to prevent the condition. Firstly, this paper constructs the relationship between behavioral characteristics (roaming), pathological characteristics (cognitive impairment), and environmental factors interacting with cognitive disorder patients. Second, the feasibility of spatial design to assist the rehabilitation of the condition was demonstrated by analyzing spatial accessibility, visual visibility, and walking distance using the environmental modification of Yuexiu Elderly Service Center in Guangzhou as an example. The study concludes that environmental influences to improve the quality of rehabilitation and cognitive function are effective, mainly in the improvement of spatial communication depth and care efficiency variables.

**Keywords:** pathological characteristics; cognitive disorders; environmental adaptations; behavioral characteristics; rehabilitation spaces; design methods

## 1. Introduction

Environmental healing has been given a spiritual element in modern times. Chinese researchers favor more active patient engagement and therapy [1], in contrast to Western scholars who stress passive treatment in space and feel that spatial areas have a healing character [2]. When considered together, research on how the environment affects people with cognitive problems has progressed from focusing on experience to meeting requirements to perceived stimulation to cognitive rehabilitation. In reality, from a single discipline perspective, it is challenging to scientifically suggest spatial design methods that are advantageous to the rehabilitation of such patients [3] due to the complexity of the processes of action of rehabilitation environments [4]. A large body of research work has led us to revisit certain environmental factors that influence the rehabilitation of this

patient population—the environmental adaptation aspects of the physical properties and pathological characteristics of space. Due to the large number of factors affecting the quality of recovery, this paper focuses on the spatial layout aspects and the process of reasoning is experimental in nature.

Worldwide, there were more than 55 million people with cognitive problems in 2021 [5]; this number is expected to rise to 131.5 million by 2050, growing at a rate of around one person every 3 s and 10 million cases per year. The therapeutic requirements of this group of individuals can no longer be met by the conventional medical approach. Due to the labor-intensive nature of non-pharmacological care interventions, the high cost of pharmaceutical treatment [6], and the possibility of unanticipated side effects, it is not universally used [7]. Most families hide their condition or even forgo treatment due to caregiving obligations and financial constraints [8]. Families demand more accessible, affordable, and individualized treatment choices, but 65% of caregivers say they do not see any chance for recovery [9]. The hypothesis that the environment facilitates recovery was proposed early on, and in 1983 Professor Ulrich provided rigorous and valid clinical data to support it, confirming the close relationship between the efficiency of recovery and the physical environment [2]. Assisting patients to heal through subtle environmental influences has prospective advantages.

In related theoretical research, Professor Ulrich put forth the "stress recovery theory" [2], which contends that exposure to or entry into a natural setting while under stressful circumstances can quickly promote physiological recovery and relaxation, and that positive feelings are typically reflected in improved functionality. Studies have shown that patients who heal in a properly designed natural environment do better in psychological adjustment and physical recovery than in an undesigned natural environment [10]. According to Kaplan's theory of attentional recovery, the recovery of directed attention is accomplished by switching attention in the natural world from directed to undirected [11]. The space should be designed in such a way that the patient's attention is actively shifted. The "sensory stimulation" idea, which has three stages, was put out by Guo Tinghong [1]. The first level is the "natural benefit," or the stimulation of the patient's senses through the amplification of the environment. According to Gesler, both man-made and natural surroundings may help with the restoration and maintenance of one's physical and mental health [12].

Environmental interventions are often accomplished in terms of affecting user behavior, i.e., the primary influence of space on human behavior, in terms of relevant practical studies. The organism's reaction to environmental changes is an adaptive activity, which Clare extends to the level of cognition by taking into account the realization of perception as a process of mental environment processing [13]. Neal Martin used "reminiscence treatment" to implant old pictures throughout patients' environments, which eventually helped the patients' cognitive performance [14]. According to Chang Huaisheng, behavioral representations reflect the extent to which organisms perceive spatial scenarios [15]. By displaying cherished items in the rooms of cognitive patients, Selwyn Laurie has been able to delay cognitive deterioration and lower depression [16]. Studies have shown that meeting psychological requirements may have a significant impact on behavior. By creating a circular walking space while also meeting patients' wandering needs [17], these studies successfully decreased the likelihood of wandering and becoming lost in cognitive patients [18].

It is clear that recent examples of research have revealed that the improvement of rehabilitation efficiency can be achieved through environmental interventions, as reflected in the qualitative and quantitative aspects affecting the information exchanged by people in space. Spatial affinity, which is closely connected to spatial layout, can be used to regulate the interchange of information. For instance, Gary discovered that the interaction that resulted from seniors sharing coffee pots to one another in a retirement community delayed cognitive deterioration in the elderly [19]. Somerset found that individuals with cognitive problems quadrupled the length of continuous discussion when the seating arrangement in

hospital activity areas was modified from row to row to centripetal. According to Chunlei Guo's research, relocating the nurses' station to the center of the facility to create a public gathering place and improving spatial communication helped patients become more involved in their surroundings [20]. Human behavior is guided and constrained by the physical environment, but this is not the only factor. A component that prior research has a tendency to disregard, according to Studer, is "factors influencing a person's behavior also include his hereditary features [21]." Some academics, including Hamilton [22], Jin Huxian [23], Wu Fan [24], Su Xuechen [25], and Roberts [26], have prioritized the environmental adaptation component of patients' pathological characteristics, favoring theoretical studies that still require expansion in terms of the strength of their justification for real-world scenarios.

The design of spatial layouts that are advantageous for the rehabilitation of people with cognitive problems are examined in this research. Cognitive problems are among the most widespread and common elderly diseases, typical in nature, and lacking entirely curative medications [27]. Physical aging is often accompanied by a variety of geriatric diseases. Delaying the onset of the condition is its most direct and effective method, and the proposed MCI stage offers a window of opportunity for spatial interventions for the rehabilitation of patients with cognitive disorders [28]. Additionally, the study presented in this paper demonstrates the important role that rehabilitation space research plays in practice for populations that are similar to those studied. Therefore, the concept of related terms needs to be defined, and the term "cognitive patients" in this paper refers to normal elderly, MCI stage, and mild cognitive patients, as moderate and severe patients are minimally influenced by the external environment. The limitation of this paper is that it does not include more influencing factors than the spatial layout, but it also makes the findings more focused on the physical properties of the space itself. In fact, the key to rehabilitation space creation is to fully understand the study population.

This work evaluates the pathological traits of patients with cognitive disorders and links them with the layout environment using the sociological research method of "environment-behavior-neuroscience [29]". Second, we match the theoretical underpinnings of spatial design from various disciplines to aid in the rehabilitation of cognitive patients, using the Yuexiu Elderly Service Center in Guangzhou as an example. Finally, we use a variety of tools to show the viability of spatial layout interventions to aid in the condition's rehabilitation. The efficacy of the environment before and after optimization is then evaluated and tested using quantitative data. The last step is a summary of the spatial layout design techniques that aid in patients' rehabilitation. Their applicability, generalizability, and practice variability are explored, and practical solution avenues are suggested for research of a similar kind.

## 2. Research Methodology and Site Overview

### 2.1. Research Subjects

Cognitive disorder is a primary progressive disease of the central nervous system characterized by cognitive dysfunction and memory impairment [30]. The three phases of cognitive impairment are mild, moderate, and severe; individuals in the severe stage will eventually acquire aphasia, dyscognition, and disuse, with a poor possibility of interventional therapy. Patients who fall into the MCI group have some degree of cognitive impairment but nevertheless preserve their ability to do daily tasks. MCI is a potentially transforming condition before being diagnosed with moderate dementia [28].

### 2.2. Research Method

Hamilton said that using a multidisciplinary study strategy is more logical and that both environmental and non-environmental elements may affect a person's behavior [4]. In order to achieve this, he developed the "environment–behavior–neuroscience" trinity sociological research method (Table 1). Its general steps are as follows: (1) thoroughly understand the subject's behavior using a variety of techniques, such as behavioral obser-

vation and expert consultation; (2) collect indicators of the patient's health variables using instruments and create a link between these indicators and behavioral variables.

**Table 1.** A trinity of environmental–behavioral-neuroscientific model of sociality research.

| Environment-Behavior | | Neuroscience | | Design |
|---|---|---|---|---|
| **Variables, Research Methods and Techniques in Various Fields** | | | | |
| Behavioral outcomes | Performance Results | Neuroscience Factors | Physiological factors | Physical environment elements |
| Observation method, photo documentation, self-reporting, etc. | Clinical records, performances, expert evaluations, etc. | PET scan, MRI, ERP evoked potentials, etc. | Testing physiological responses, such as cortisol testing, blood pressure testing | Describe environment specific Features, such as layout, scale, etc. |

Some of the phases in the research technique may not apply to this study because it was designed for the medical profession. In accordance with its logic, the author transforms the research arrangement into (1) an understanding of the research object through literature review, expert consultation, and observation method, and construction of the correlation between the object attributes and the environment; and (2) a discussion of the viability of spatial design intervention in the rehabilitation of the condition in the context of the pathological characteristics of patients with cognitive disorders; (3) replacing the evaluation index from medical variables to environmental variables, assessing the effectiveness of environmental modification and summarizing the design methods. This research is a thorough study built on the integration of several theories, methodologies, and instruments due to the intricacy of the impact of rehabilitation quality and environmental elements [29,31] (Table 2). With the aid of Stata statistical charts, this study will employ Depthmap and GIS for geographical accessibility analysis in the feasibility phase and Ecotect for visibility analysis in the assessment phase.

**Table 2.** Environmental factors affecting the quality of rehabilitation.

| Medical Effects | Environmental Influences | | | | | | | |
|---|---|---|---|---|---|---|---|---|
| | Single Room | Natural Light | Artificial Lighting | Natural Scenery | Flooring Material | Reduce Noise | Rational Layout | Distracting Patients |
| Reduce getting lost | | | • | | | • | •• | |
| Reduce medical errors | • | | • | | | • | | |
| Reduce patient falls | • | | • | | • | | | |
| Reduce pain feeling | | •• | • | •• | | • | • | •• |
| Improve sleep quality | •• | • | • | | | • | | |
| Reduce patient stress | • | • | • | •• | | •• | | |
| Reduce negative emotions | | •• | •• | • | | | | |
| Reduce waiting time | | • | • | • | | | | |
| Improve privacy and convenience | •• | | | | | • | | |
| Enhance communication | •• | | | | | • | | |
| Add satisfaction | •• | • | • | • | • | • | | |
| Reduce the stress of nursing | • | • | • | • | | • | | |
| Improve healthcare efficiency | • | | • | | | • | •• | |

• The two are directly or indirectly linked. •• Both have a strong or supportive relationship.

In the presentation and assessment phase of this work, connectivity is a significant environmental variable that is mostly gained through simulation using the spatial syntax program Depthmap. The total amount of places that are directly connected to the spaces around them is what is known as connectivity. A space is considered to be a transportation hub in the spatial system if its connectivity value is high, because it is connected to more of other locations. Connectivity can also reveal an observer's field of vision inside a certain area. Equation $C_i = k$ yields this result, where $k$ is the number of nodes that are directly linked to the i-th node.

## 2.3. Site Overview

Yuexiu Elderly Service Center in Guangzhou is the first cognitive day care zone in Guangzhou, with a total indoor area of about 2006 m$^2$ (five floors in total), a total of 21 faculty members, about 3000 registered members, and about 300 active elderly persons. The fourth floor is the cognitive care zone, with an area of about 660 m$^2$, and the service targets are mainly MCI and mild cognitive patients. This paper takes the spatial layout transformation of this floor as an example for in-depth study. The main space of this floor is divided into spring dawn garden, calm the mind, dining Area, bathing room, nurses station, etc. The nurses' station is set to the left of the entrance (Figure 1), and the overall decoration style is "family style", with wood as the main finish material.

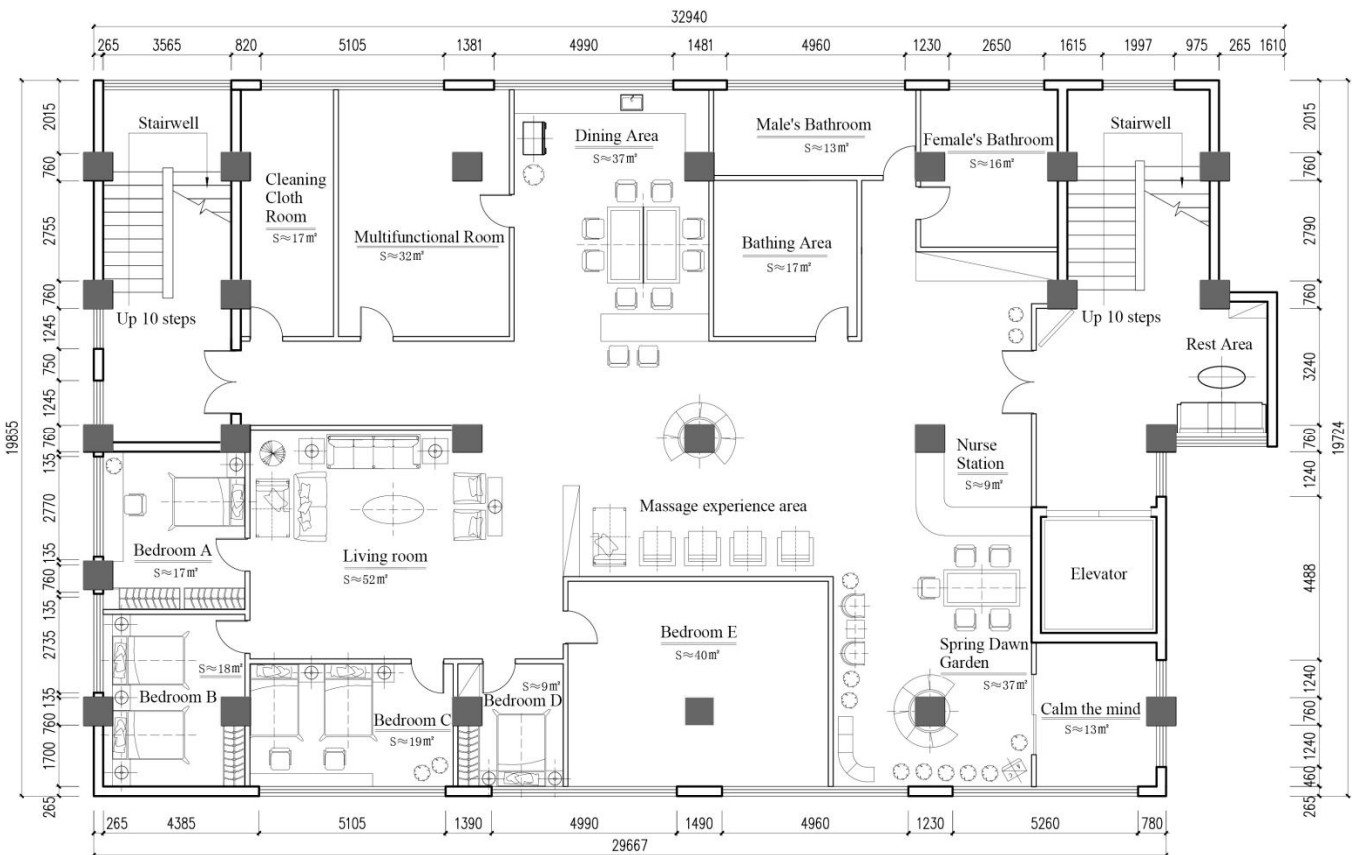

**Figure 1.** Floor plan of the fourth floor of Guangzhou Yuexiu Senior Citizen Service Center.

## 3. Feasibility of Spatial Design Intervention in the Rehabilitation of Cognitive Disorders

### 3.1. Relevance of Disease Recovery to the Environment

#### 3.1.1. Cognitive Impairment and Spatial Layout

The human cognitive system is closely related to space [32]; improving the spatial environment might delay cognitive aging [26,33]. Wandering behavior is a common symptom of patients with cognitive impairments [34] and is positively connected with the severity of the impairment [35]. Sedentary behavior in patients with cognitive disorders has also been suggested as a potential risk factor for exacerbating cognitive impairment [36]. Behavioral and psychological symptoms of dementia (BPSD) "demand-driven compromise behavior" [37] consider the abnormal behavior of this group of patients as an outward expression of their unmet internal needs [38]. The tracking of human activity patterns in space may result in changes to the physical environment. For instance, the Nuremberg Dementia Care Center in Germany designed a circular path for patients to wander in circles [17], interspersed interesting functional partitions at each path node, and created a high level of sensory stimulation during patients' wandering, distracting them and sparking their desire

to engage in activities, increasing their communication with the environment and thereby reducing meaningless wandering behaviors. Related studies have demonstrated the strong correlation between roaming behavior and environmental factors in patients with cognitive disorders (Table 3).

Even basic body language, such as eye motions, welcomes, and head nods, can stimulate brain activity and decrease cognitive loss. Bathrooms near to windows were observed to have an average increase of 1.3 spatial communication [39], and the depth of spatial communication was strongly connected with floor layout [40].

**Table 3.** The association between roaming behavior and environment in cognitive patients.

| Author | Opinions or Research Findings |
| --- | --- |
| Yanan, W. [17] | At the Nuremberg Dementia Care Center in Germany, patients' interaction with their environment is increased by creating pathways that facilitate wandering in circles and rich environmental stimuli, thereby reducing meaningless wandering behavior. |
| Backhouse, T. [41] | Wandering may be more of a problem for patients in home care than in institutional care settings. |
| Algase, D L. [42] | Wandering behavior overlaps with the spatial orientation problem, which is closely related to the spatial context of wandering. |
| Coons, D. [43] | However, the environmental context lacks attraction, engagement, and belonging, and spontaneous activities of restless wandering may occur. A stimulation-rich environment can reduce wandering behavior. |
| Snyder, LH. [44] | Goal orientation, stress, and boredom are the three main causes of wandering behavior. However, when the environment is unattractive, it may increase the wandering behavior of patients. |
| Calkins, M. [45] | As cognitive decline blurs the familiarity of the environment for people with cognitive disorders, unfamiliarity can become frightening and may prompt continuous wandering behavior. |
| Dunkle, R.E. [46] | The stressors associated with moving to a new environment may increase wandering behavior. |
| Dickinson, J. [47] | Wandering behavior of vagrants may be attributed to boredom or stress. |
| Hussain, R.A. [48] | An interesting environment can distract the patient, thus reducing the boredom and restlessness of wandering. |
| Calkins, M.P. [49] | Interactive artwork is used along a corridor in a dementia care unit at the Iowa Veterans Home in Iowa City, and it can interrupt the patient's wandering behavior. Although simple in design, it adds visual interest to the unit and, more importantly, it may have therapeutic value when wanderers stop and interact. |

### 3.1.2. Nursing Efficiency and Line of Sight Access

According to the Society of Critical Care Medicine's design recommendations, there should be a clear line of sight between the patient and the nurse's station [50]. High visual accessibility fosters cooperation and communication [51], shortens walking distances, and enhances the effectiveness of treatment. It also enhances the effectiveness of emergency patient care [52]. The layout of a floor plan directly affects how easily people can see space [53]. Modern hospitals frequently adopt the cross-shaped ward design because it makes it easier to observe patient behavior and vastly increases the effectiveness of nursing care. A high degree of sight between departments is provided by the circular corridor plan, and the larger the corridor, the better the visibility [54]. According to the Chinese Institute of Architecture, a space's recognizability may be improved by optimizing the layout's orientation [55].

### 3.2. Feasibility of Spatial Design Interventions for the Rehabilitation of Patients' Conditions
### 3.2.1. Adjusting Spatial Layout to Improve Cognitive Function

Previous research has demonstrated a strong correlation between individuals with cognitive disorders' wandering behavior and cognitive performance. The concept supporting wandering was proposed: "It can be a means for wanderers to exercise [56], can minimize problem behaviors, and can promote sleep after physical activity [57,58]". Physical constraint against wanderers has been shown to increase the risk of behavioral issues. Rocha [59] and the Chinese Special Committee on Microcirculatory Neurodegenerative

Diseases [60] both agreed with this viewpoint. This theory that wandering behavior may be decreased is supported by related studies, and several recent design examples are beginning to create roaming trails for wanderers [17].

A spatial experiment was carried out in Japan to look at wanderers' wandering behaviors [61], and the author contributed and deduced from this experiment (Figure 2) that patients with cognitive disorders tend to wander aimlessly in empty rooms, but when there are clear indicators, they are drawn to move closer to the target. Therefore, it stands to reason that by strategically putting markers, patients might be directed to roam. Patients' memory sequences are arranged sequentially at various spatial nodes with directional paving materials and colors to let them wander along the designated routes and help them build associations in their memories while they roam. This approach can direct the link while it roams, contribute to brain activation and improved cognition, and lessen the likelihood of disorientation.

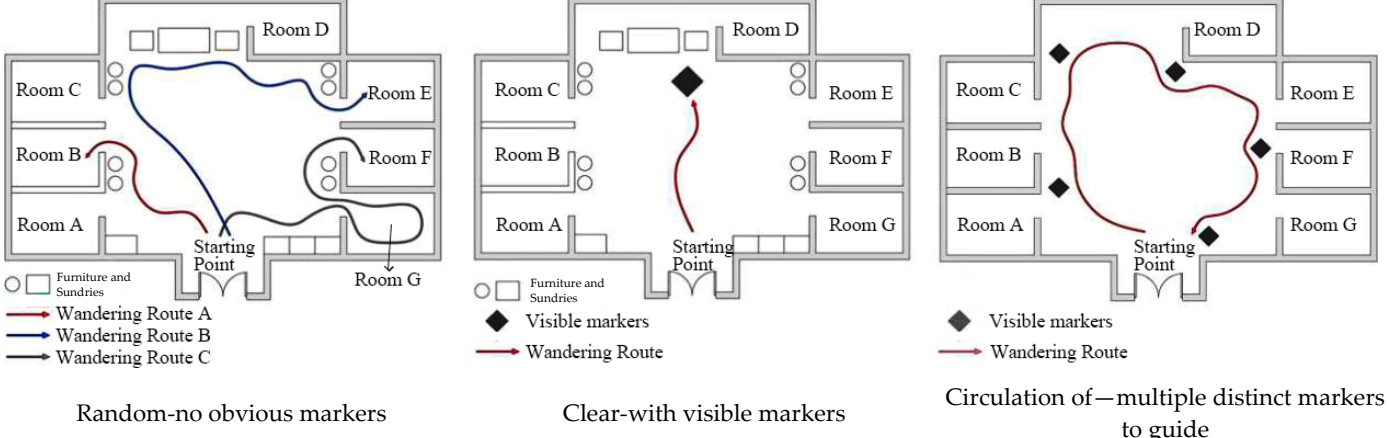

**Figure 2.** Spatial cognition experiment in cognitive patients.

On the basis of this approach, the author modified the single-corridor layout of the cognitive care area to a multi-core layout (Figure 3), forming four short-circulation areas for patients to roam with the nurse's station as the core and interspersing interesting spaces in the roaming paths to encourage regional composite functions and improve spatial connectivity to increase walking appeal [62]. In order to improve the depth of spatial communication, roamers are directed to gather in a certain location.

Patients with cognitive impairment frequently wander aimlessly. In this study, we simulate random human flow paths using the Depthmap tool and assess connectedness before and after layout modification. The spatial connectivity (noted as Ci) is calculated from the chosen entrance point as the beginning point, and the intensity of the grammar computation is visually shown in three levels and nine levels. A grid line with Space = 650 mm (near to the real scale) is established in Set Grid (Figure 4). According to the simulation (Figure 5), the total spatial connectivity is often at a medium level ($4 \leq Ci \leq 6$) before the layout modification, and it is at a low level between the two stairs ($1 \leq Ci \leq 2$). The simulation in Figure 6 shows that the overall attraction of the space is uneven. The living room, central area, massage experience area, foyer, and spring garden area all have Ci values that are higher and, on average, in the medium level (Ci = 5), while the other areas are primarily in the low level (Ci = 3). The circular region created by the massage experience area had greater connectivity in the two simulations, and it may conceivably be utilized as a small-scale wandering path for people with cognitive problems.

The open space was created to improve the depth of communication after the nurses' station was moved to the center and the partition wall between the living room and the multi-functional room was opened (Figure 7). Additionally, the overall spatial connectivity broke the tendency of "one" shape, which is not suitable for the convalescent space, and the radial form was formed, forming a multi-circulation path layout with the nurses' station as

the core (Figure 8). The multi-purpose room and cleaning room were moved, the cleaning area was separated from the cleaning room, the eating area was expanded in size and usefulness, and the living room and nurses' station were linked to create a clear line of sight communication. The simulation's findings (Figure 9) indicate that the living room's connectivity increased from $C_i = 5$ to $C_i = 6$. It is noteworthy that the local space now has much more visual connectivity, and the entryway and living room area were raised from the medium level ($C_i = 6$) to the high level ($C_i = 8$).

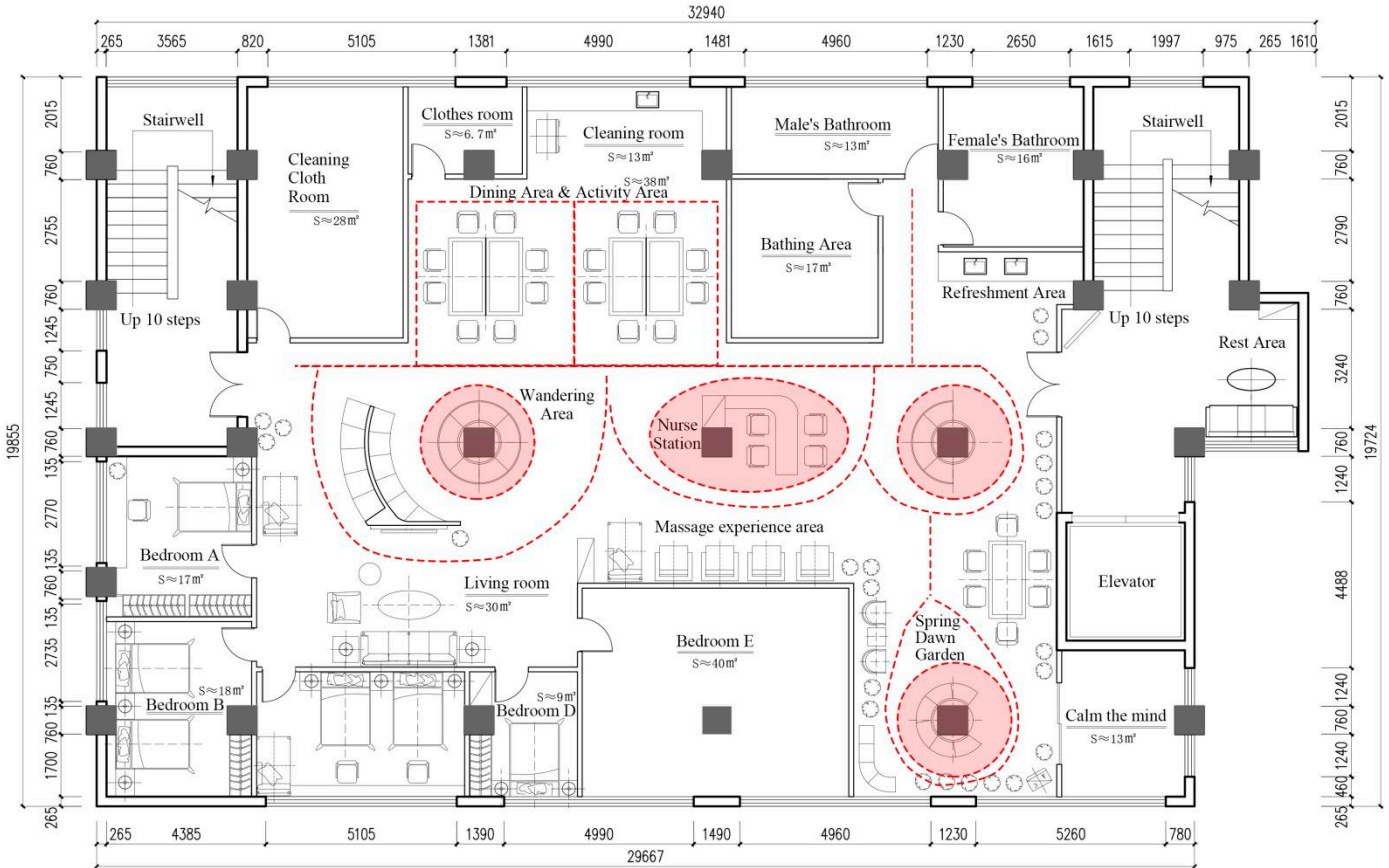

**Figure 3.** Plan after renovation.

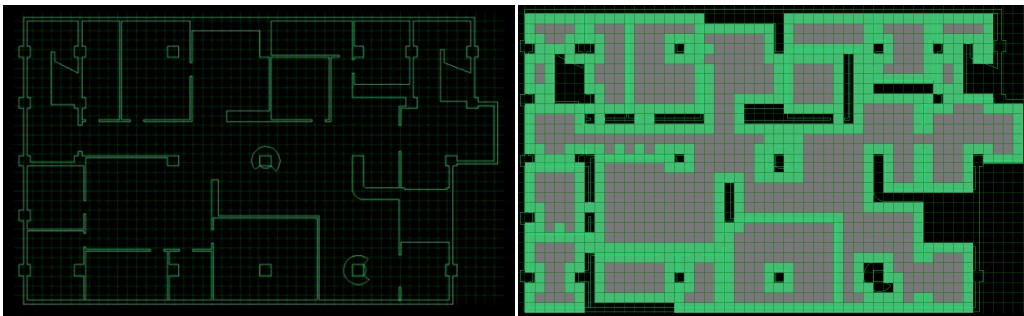

**Figure 4.** 650 mm grid line and calculation range setting.

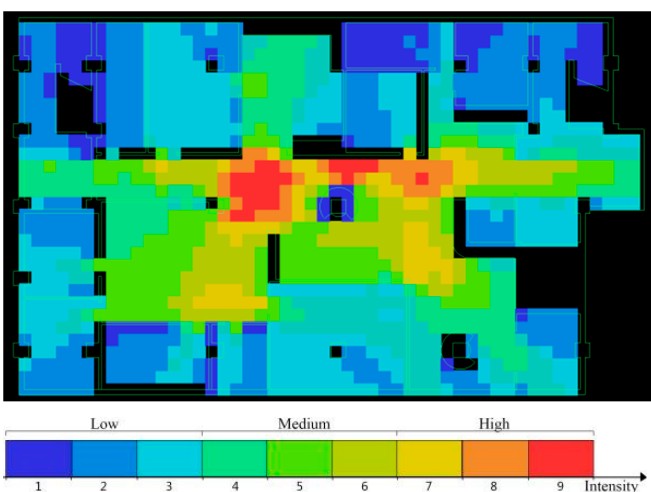

**Figure 5.** Spatial accessibility before layout adjustment.

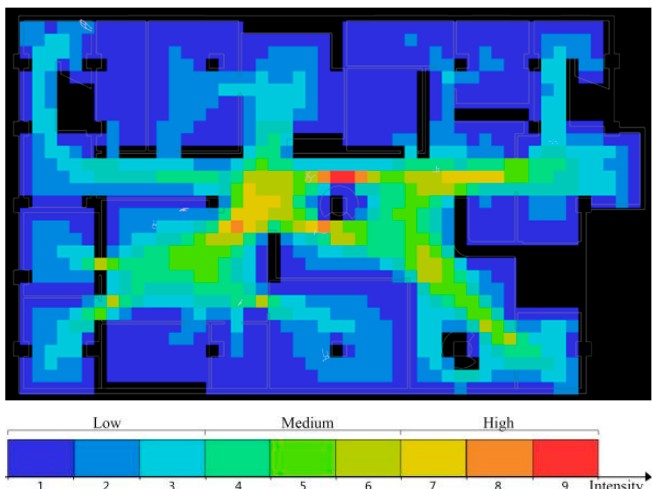

**Figure 6.** Spatial attractiveness before layout adjustment.

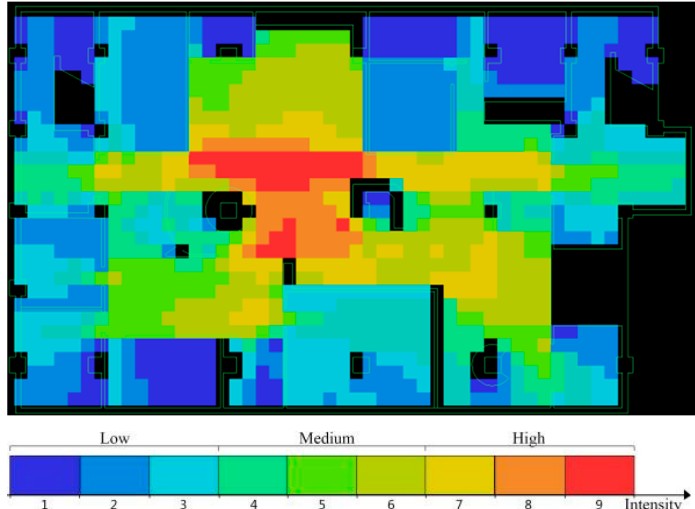

**Figure 7.** Spatial accessibility after layout adjustment.

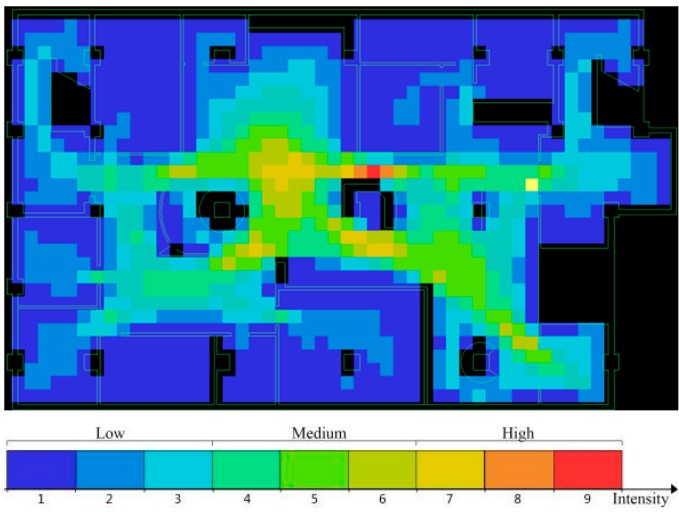

**Figure 8.** Spatial attractiveness after layout adjustment.

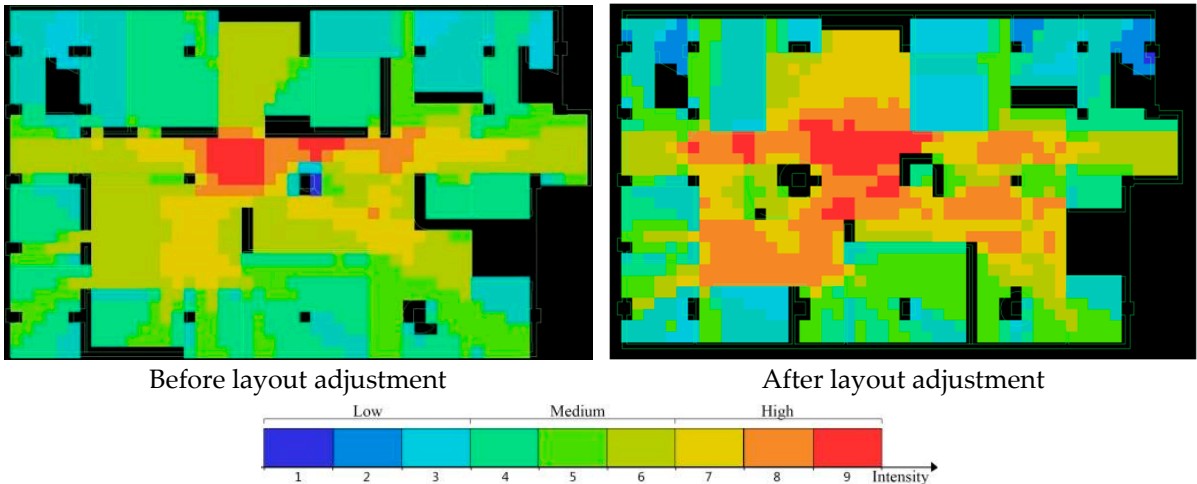

Before layout adjustment | After layout adjustment

**Figure 9.** Perspective story trigger levels.

### 3.2.2. Optimize Visual Access to Improve Care Efficiency

Daily safety concerns are a top worry for families due to the physical and psychological dysfunction of people with cognitive impairments [63]. According to a study [64], people with cognitive problems who were missing for more than 24 h had a death rate as high as 46%. A high degree of visibility can enhance nurses' ability to watch patients [51]. The level of spatial visibility is connected to ward layout, bed orientation, and the position of open doors and windows. The ward's layout should be thought up with the patient's primary perspective in mind while they are laying in bed, with a side view out the window being ideal since it enhances the patient's exposure to nature.

Further modifications were made to the bed and door opening locations based on the optimum arrangement, and simulations comparing the three alterations were run (Figure 10). The major purpose of changing the bed's position was to give the patient's eyes more time to contact the window while they were lying in bed. The major reason for changing the opening positions of the doors to bedrooms A and C was to provide direct line-of-sight contact between the nurse and the patient in the out-of-room pathway.

The simulation comparison revealed (Figure 11) that the door opening and ward locations were changed to allow the nurses along the route to see the bed position clearly. The ease of sight reach is assessed using sight integration [65], with red spots denoting strong integration and increased visibility. These simulation results led to the selection of two distinct views with good visibility levels for line of sight analysis in GIS. The results

show (Figure 12) that before the layout adjustment, points A1 and A2 were more restricted in view acquisition due to narrow space and wall obstruction, and the visual visibility was significantly improved after the layout adjustment (point B1), but the visual penetration level of points B1 and B2 into the ward was still unsatisfactory, while the range of visible values increased (green line) and the visibility level improved (points C1 and C2) after adjusting the position of beds and door opening.

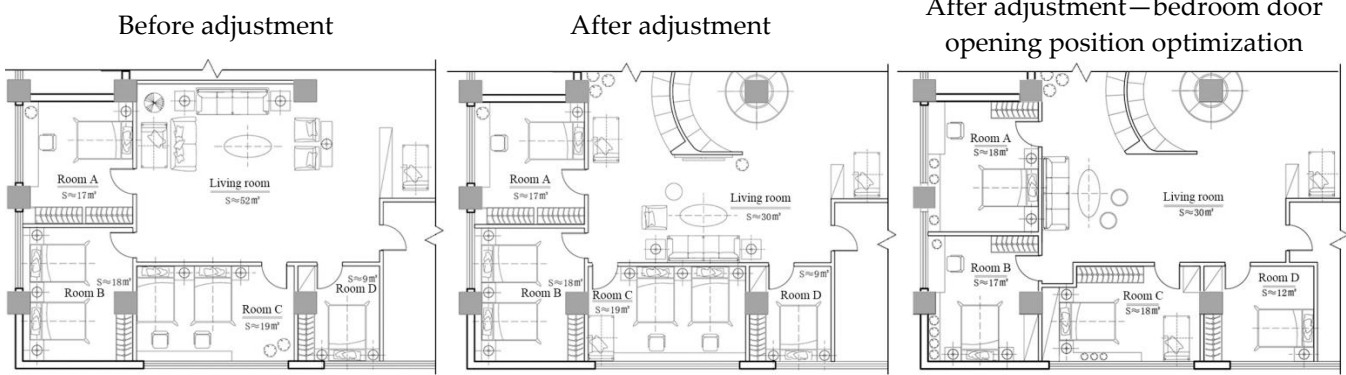

**Figure 10.** Comparison of the details before and after the layout optimization.

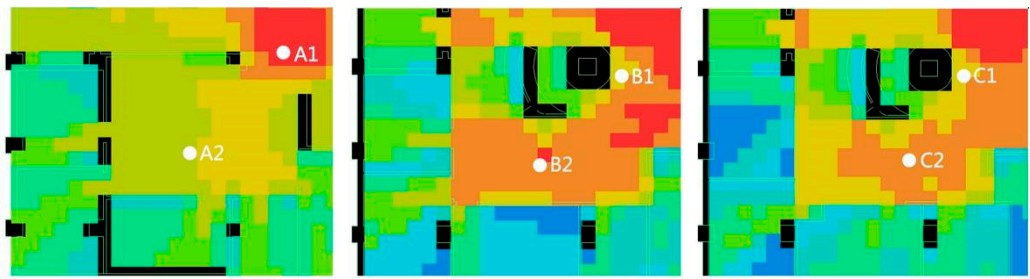

**Figure 11.** Comparison of visual accessibility before and after layout optimization.

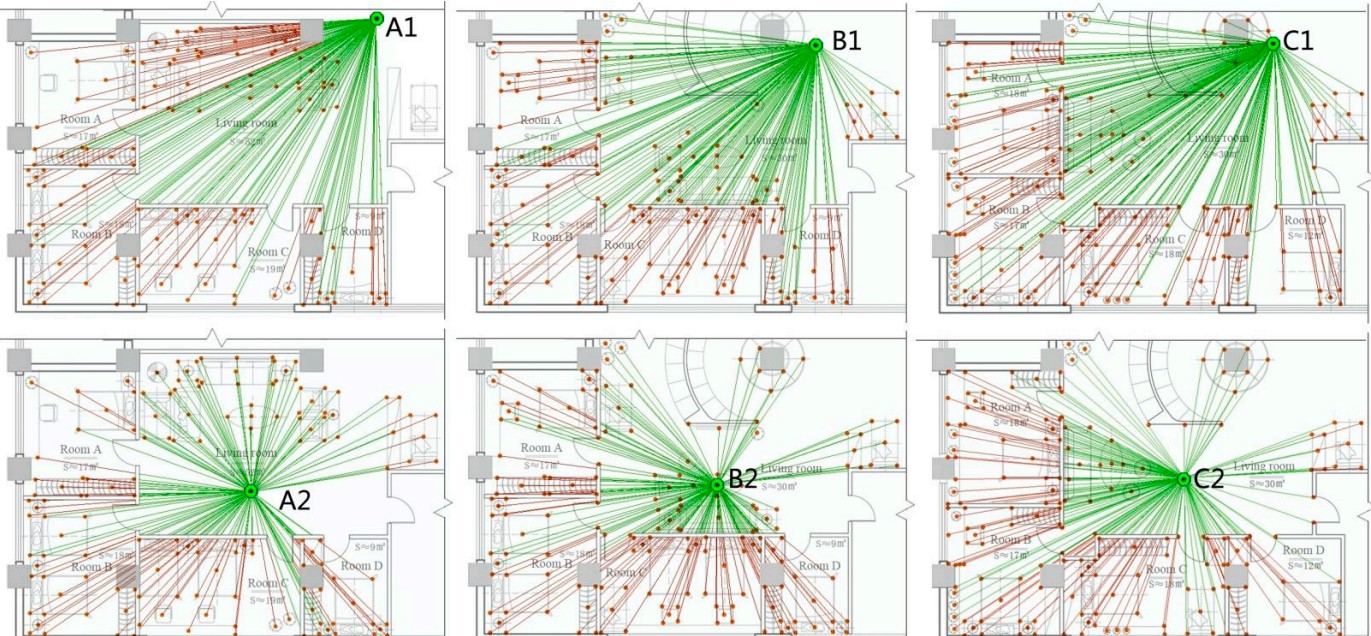

**Figure 12.** Through-view analysis of different viewpoints (green is visible, red is invisible).

## 4. Space Transformation Optimization Assessment

This article is separated into two parts of assessment: visual field visibility and care efficiency, for the layout optimization of the cognitive care area. The ECOTECT software's built-in features were used to analyze visual field visibility, which was then assessed by the change in the visual field visibility value, with a higher number suggesting easier observation by other spaces. In the United States, the effectiveness of hospital rounds is now gauged by how far it is from the nurse's station to the ward [66]. As a result, the shortest linear distance that can be covered on foot from the nurses' station to each functional area serves as the primary evaluation criterion in this paper's assessment of care efficiency. This value was calculated using CAD software. According to related academics, the distance between the nursing center and the farthest ward should not be greater than 30 m [67].

### 4.1. Visual Visibility

ECOTECT examined the visual field visibility before and after the layout alteration. The results show (Figure 13) that the area of the yellow zone is significantly increased; the visual field visibility is significantly improved after the layout adjustment; and the visibility levels of the nurse's station, public area, and bedrooms A to D are all improved overall, with the visibility levels of bedrooms A and B increasing from the blue zone value to the red zone value.

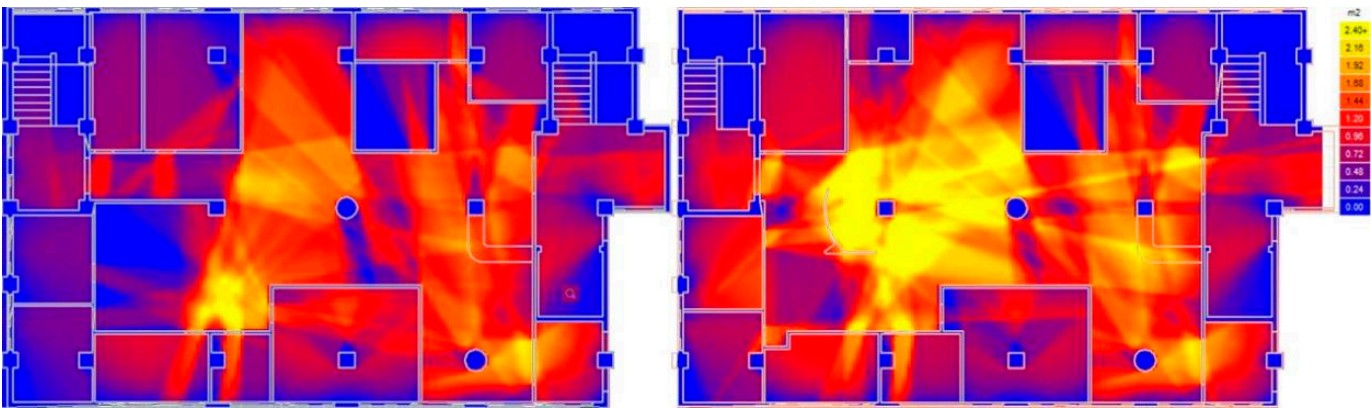

**Figure 13.** Comparison of field of view visibility (**left** is before adjustment, **right** is after).

The field of view exposure shows the extent to which it can be observed by other spaces. The visualization data was exported via ECOTECT (Figure 14) and plotted biaxially, and then the peaks on each axis before and after the layout adjustment were extracted (Figure 15). After the layout optimization, the view exposure of most areas was significantly improved, including bedroom A from 0.65 m$^2$ to 1.84 m$^2$, corridor from 1.85 m$^2$ to 2.09 m$^2$, living room from 2.22 m$^2$ to 2.83 m$^2$ (now adjusted to public space), central area from 1.85 m$^2$ to 2.05 m$^2$ (now nurse station location), former nurse station area from 1.48 m$^2$ to 1.73 m$^2$, spring dawn garden from 1.61 m$^2$ to 2.07 m$^2$, the entrance from 1.80 m$^2$ to 1.98 m$^2$, the observable level of the stairwell near the bedroom A side was also significantly improved (from the peak 0.58 m$^2$ to 1 m$^2$), and the exposure rate of the view between the Calm the Mind space remained almost the same.

On the basis of Figure 15, the optimal field of view exposure region was further extracted to generate Figure 16. After the modification, the best view region of the public space grew from 3 (points A1, A2, and A3) to 6 (points A1–A3, B1–B3), and the best view point of the bedroom climbed from 1 (point B1) to 3 (points B1–B3). The dining area, living room, and nurses' station were connected to form a public space with high interactivity—the A4 area—as well as forming a view area (C area) for simple observation of daily activities of cognitive patients. The original layout was a more closed space with

a greater degree of separation between areas, which could not form a more concentrated observable area.

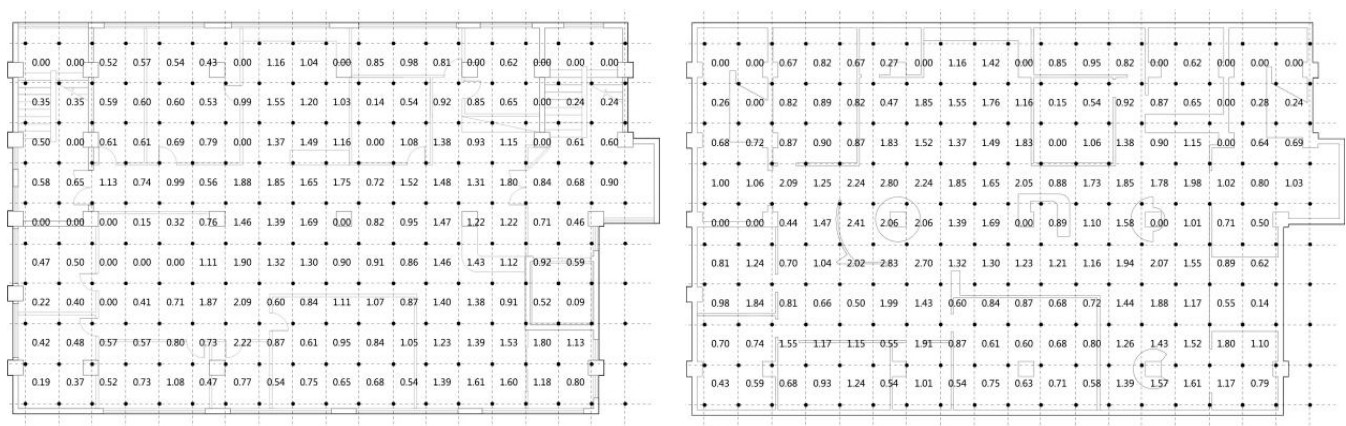

**Figure 14.** Comparison of field of view exposure (**left** is before adjustment, **right** is after).

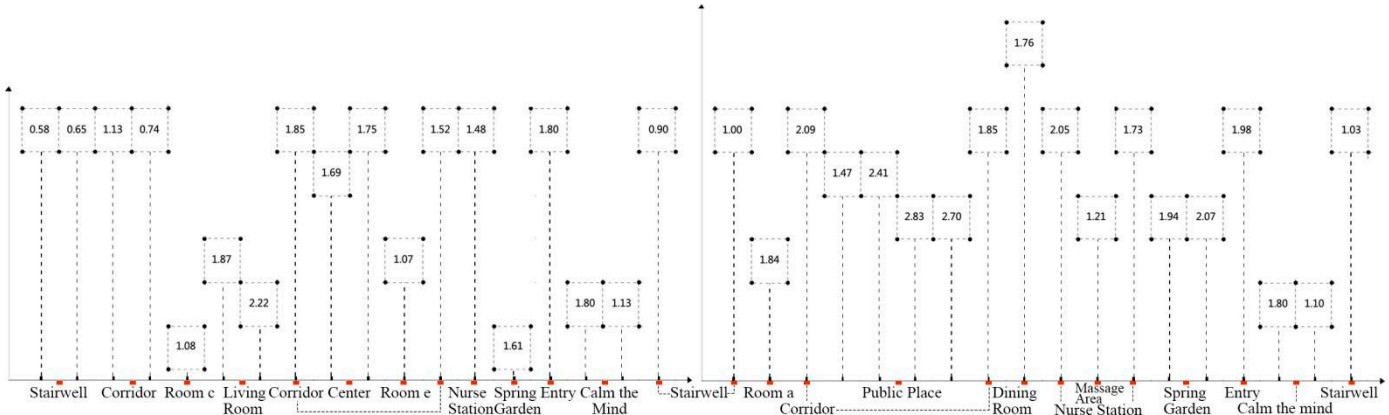

**Figure 15.** Peak area on the 2D axis (**left** is before adjustment, **right** is after).

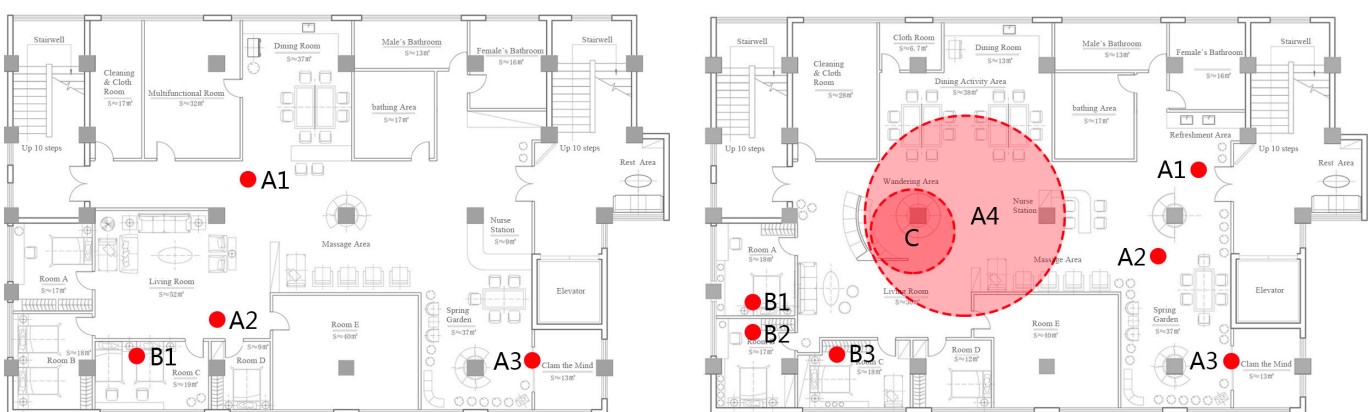

**Figure 16.** Number of optimal viewpoints and regional variations (**left** is before adjustment, **right** is after).

### 4.2. Nursing Efficiency

At least 40% of a nurse's day is spent on foot [68], and how much time is spent walking depends on the layout, size, and visibility of the area. Foot distance is also impacted by complex work flows, and if a ward's entrance size is extended from 3.6 to 4 m, the daily maximum walking distance will rise by 1000 m [69].

Peter Manning estimated the spatial gravitational intensity in terms of the topological complementary distance, based on which a feasible plan layout was deduced, and then split a floor of the inpatient department into a location matrix based on the column network structure. In accordance with this theory, this floor was divided into a gravitational network of 11 × 20 (Figure 17), the gravitational strength was coded from the nurse's station as the core to the periphery from strong to weak, and the gravitational strength and the shortest walking distance between the nurse's station and each functional area were compared (Table 4).

Following the layout change (Table 4), there was a considerable reduction in the distance between the nurses' station and the massage experience area, dining area, living room, bedroom, cleaning room, multifunctional room, and stairwell. In particular, the distance to bedroom B was cut by 24.2 m. Secondly, there is a small reduction in the distance to spring dawn garden and Calm the mind, and the reduction in the distance to the elevator, rest area, refreshment area, bathing area, male's bathroom and a stairwell varies within the range of 2.07 m to 5.8 m, and there is almost no change in the distance to the female's bathroom. The gravitational pull is the same in the nurses station, spring dawn garden, and the female's bathroom. The overall decrease in walking distance was 83.69 m.

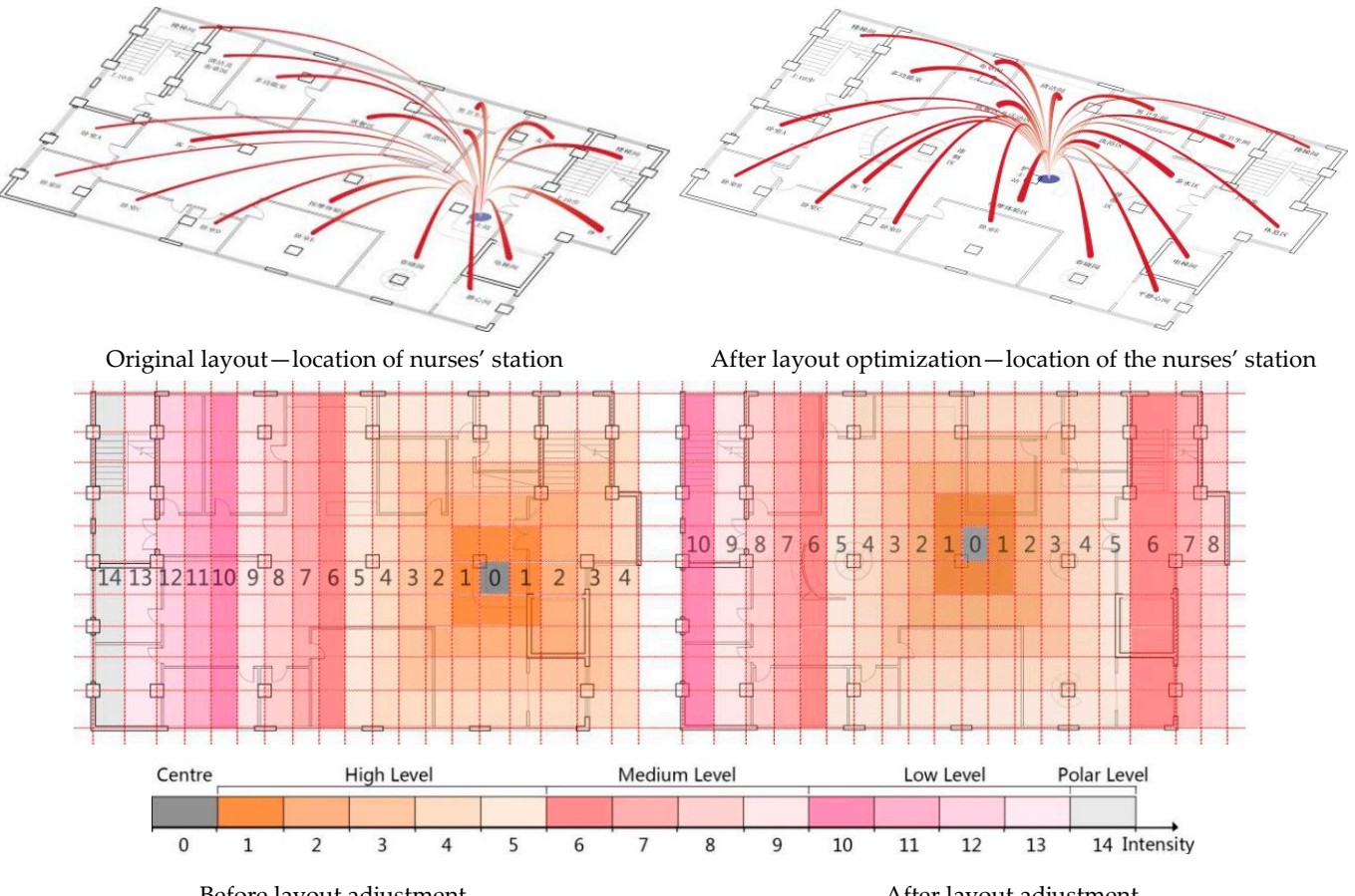

**Figure 17.** Relationship between the gravitational strength of the nurses' station and each functional area.

Table 5 shows that this floor has a total of 20 functional sections, 13 of which have walking distance reduction number and number of gravitational enhancements, with 65%. Following the layout modification, the number of high gravitational areas remained at 10, the number of areas with medium level increased from 3 to 9, the number of areas with low levels significantly decreased from 4 to 1, and the number of areas with extremely polar levels made a breakthrough from 3 to 0 (Table 6). In conclusion, the relocation of the

nurses' station to the hub area greatly cut down on the walking distance and increased care efficiency.

**Table 4.** Comparison of the shortest walking distance from the nurses' station to each function.

| Functional Area | Minimum Walking Distance/m and Gravitational Strength | | | | | Increase or Decrease/±m | |
| | Before Adjustment | Gravitational Strength | After Adjustment | Gravitational Strength | Changes | Partial | Total |
|---|---|---|---|---|---|---|---|
| Refreshment area | 3.9 | 2 | 8.3 | 4 | + | +4.4 | |
| Rest area | 6.7 | 3 | 12.2 | 7 | + | +5.5 | |
| Elevator | 7.5 | 2 | 13.3 | 6 | + | +5.8 | |
| Bathing area | 8.2 | 3 | 11.7 | 2 | − | +3.5 | |
| Massage area | 8.3 | 5 | 1 | 2 | − | −7.3 | |
| Male's bathroom | 9.6 | 5 | 11.3 | 2 | − | +1.7 | |
| Female's bathroom | 11 | 4 | 11.1 | 4 | ○ | +0.1 | |
| Spring dawn garden | 11.4 | 3 | 8 | 3 | ○ | −3.4 | |
| Dining area | 15.8 | 6 | 8.4 | 2 | − | −7.4 | |
| Calm the mind | 17 | 4 | 14.7 | 6 | + | −2.3 | |
| Living room | 19.8 | 10 | 9.2 | 6 | − | −10.6 | −83.69 |
| Multifunctional room | 21.2 | 10 | 16.3 | 7 | − | −4.9 | |
| Cleaning room | 23.5 | 12 | 15.2 | 5 | − | −8.3 | |
| Bedroom A | 26.2 | 14 | 17.6 | 9 | − | −8.6 | |
| Bedroom B | 44.1 | 14 | 19.9 | 9 | − | −24.2 | |
| Bedroom C | 23.8 | 11 | 19 | 7 | − | −4.8 | |
| Bedroom D | 22.1 | 8 | 12.4 | 4 | − | −9.7 | |
| Bedroom E | 22.2 | 6 | 13.3 | 3 | − | −8.9 | |
| Stairwell 1 | 10.2 | 5 | 12.27 | 6 | + | +2.07 | |
| Stairwell 2 | 29.3 | 14 | 23.24 | 10 | − | −6.06 | |

Noted: + and − indicate the rise and decrease in gravitational strength respectively, ○ indicates no change.

**Table 5.** Statistics of the number of gravitational intensity regions.

| Total Number of Functional Areas/pc | Walking Distance Reduction Number/pc | Proportion/% | Number of Gravitational Enhancements/pc | Proportion/% |
|---|---|---|---|---|
| 20 | 13 | 65% | 13 | 65% |

**Table 6.** Statistics of the number of gravitational intensity regions.

| | High Level | Medium Level | Low Level | Polar Level |
|---|---|---|---|---|
| **Before Layout Adjustment** | 10 | 3 | 4 | 3 |
| **After Layout Adjustment** | 10 | 9 | 1 | 0 |

## 5. Results and Discussion

### 5.1. A Design Approach to Space Layout for the Rehabilitation of Patients with Cognitive Disorders

5.1.1. Guiding Spatial Behavior

The evolution of hospital layout is primarily focused on reducing walking distance and increasing care efficiency [70], which is essentially a transition from a process of spatial behavior orienting to high-quality information interchange. Wrap-around layouts have a centripetal force impact (Table 7) and radial layouts have a central discrete effect, which together determine the direction of spatial behavior itself. The plan layout may be separated into centripetal and non-centripetal layouts based on the domain force. When compared to the former, the non-centripetal type's centrality-oriented function is unclear. Examples include the scattered cross layout found in protestant homes in America, the overpass

surround layout found at Japan Rainbow Hills Rehabilitation Center, and the multi-core scatter layout found at Allen Nursing Home, England.

**Table 7.** Domestic and international nursing home layout model and movement model.

| Domestic and International Nursing Home Layout Models | | | |
|---|---|---|---|
| 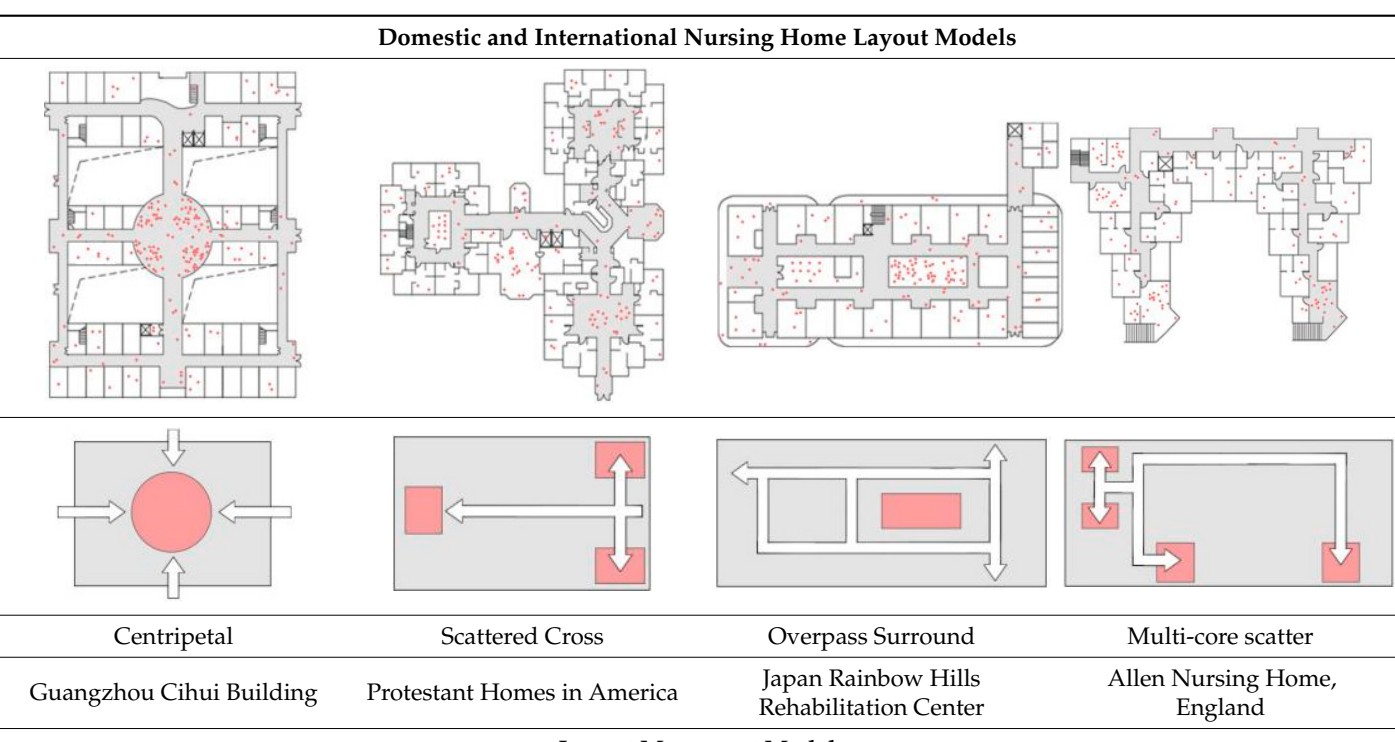 | | | |
| Centripetal | Scattered Cross | Overpass Surround | Multi-core scatter |
| Guangzhou Cihui Building | Protestant Homes in America | Japan Rainbow Hills Rehabilitation Center | Allen Nursing Home, England |
| Layout Movement Model | | | |

Matching the domain force and dominating function of the space layout with the behavioral features, pathological characteristics, and physiological characteristics of the patients is the first step in designing a rehabilitation space for people with cognitive disorders. The Guangzhou Cihui Building's centripetal architecture, for instance, demonstrates that it is more suited to the aimless roving behaviors of people with cognitive problems. This floor plan can direct patients to the center area, enabling centralized patient monitoring, increasing the likelihood of patient–doctor interactions, deepening spatial communication, stimulating brain function, and so delaying cognitive decline.

5.1.2. Reduced Walking Distance

The non-centripetal layout can be divided into two categories: matrix layout (Figure 18) and radial layout (Figure 19), the former of which includes single corridor layout and multi-corridor layout. It is best to avoid single corridor, strip, and single row layouts because their long, narrow corridors will lengthen the walking distance between patients and doctors. Comparatively, the functional arrangement is more focused, and the multi-corridor layout is superior. Even though a multi-corridor layout has a corridor that is 65% longer than a single corridor under the same functional room arrangement, the former's walking distance is 1.24 times greater. The compound layout is created on the basis of single corridor layout, such as the matrix layout "king," "field," "well," etc., which significantly enhance the visual accessibility of space. The convenience of walking deflection, visual accessibility, and spatial connectivity are all considerably enhanced by this arrangement, which also shortens the walking distance and saves time.

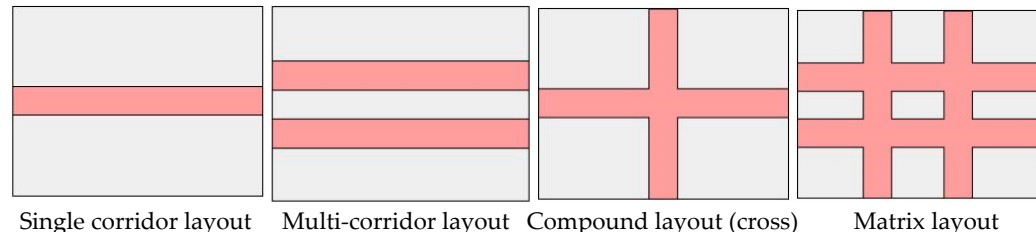

| Single corridor layout | Multi-corridor layout | Compound layout (cross) | Matrix layout |

**Figure 18.** Matrix layout pattern evolution process.

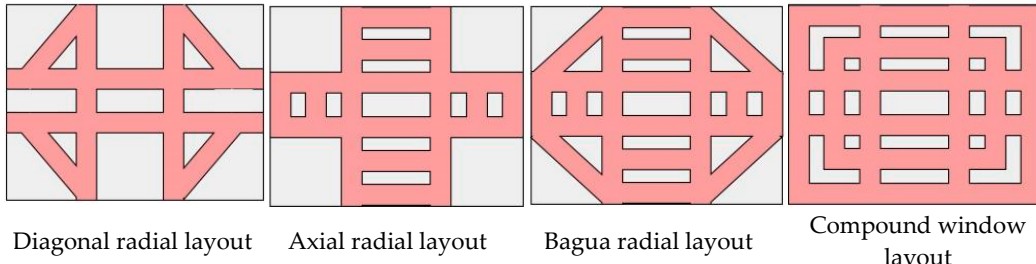

| Diagonal radial layout | Axial radial layout | Bagua radial layout | Compound window layout |

**Figure 19.** Radial layout space movement model.

The radial layout is a complex form that changes from the matrix layout by adding connecting corridors (Figure 19). Axial radial layout is based on the compound layout, which is formed by connecting corridors on the axis, with better overall functional compound and spatial connectivity than diagonal radial layout, but slightly inferior to bagua radial layout. Compound window layout is based on the matrix layout, formed by adding back to the periphery in the form of corridors; this layout has the same advantages as the bagua radial layout and can be applied to a larger area of space.

### 5.1.3. Deepening Space Exchange

The usage of the layout should match the user population's features because each layout has its own unique qualities. For cognitive patients, a centripetal layout with the central nurses' station area serving as the core and corridors added to the periphery sections to promote spatial connectivity and visual accessibility (Figure 20), establishing a radial centripetal layout, is the ideal general design. For hierarchical management in big nursing homes, additional nurse stations can be placed up at four diagonal positions, establishing a multi-core axis radial layout that works in conjunction with the central nurse station to save travel time and boost care efficiency. Additionally, based on the characteristics of cognitive patients' roaming behavior, some wandering paths should be set up in each functional area to encourage wandering. These paths can provide appropriate walking exercise on the one hand, and they can also increase the level of sensory stimulation in the area, which will help to effectively distract the patients and boost environment participation on the other. For instance, a second, smaller nurses' station may follow a circular design to provide a narrow circular circulation.

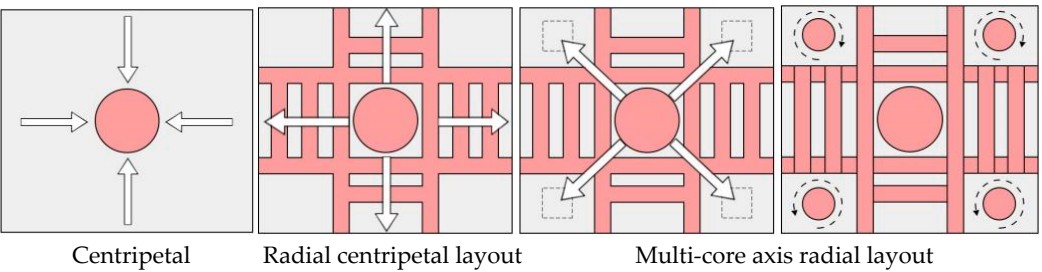

| Centripetal | Radial centripetal layout | Multi-core axis radial layout |

**Figure 20.** Multi-core axis radial layout for wandering characteristics of cognitive disorder patients.

*5.2. Discussion of Results*

In order to realize the additional objective of sustainable building realization [71] and further illuminate the prospect that environmental design might aid in rehabilitation, the research of the environmental adaptability of pathological traits is important. Regarding the methods of environmental rehabilitation, in terms of the path to environmentally beneficial recovery, the "stress recovery theory" [2] and the "attention recovery theory" [11] are implemented through the "remote viewing" path; Neal Martin [14] proposed "reminiscence therapy" (hanging old photographs in the ward), which is achieved through a pathway from remote viewing to association. The "auditory therapy" proposed by Su Xuecheng [25] is achieved through the path of listening and spiritual touch. The "aromatherapy" proposed by Hoshen Kim [23] is achieved through a pathway that affects the health of the mind and spirit. In contrast, this study provides partial data on the physical attributes of the environment (visual field visibility, visual field exposure, and walking distance) that influence the cognitive five senses experience, but more supportive data remain to be explored.

According to the degree of realization, sensory stimulation rehabilitation is divided into three levels [1]: "natural help", which amplifies space to stimulate the senses; "stimulation of thinking", which activates the nervous system by causing associations in patients; and "help for organismic developmental engagement", which uses space to direct patients to engage in activity practices and carry out a variety of tasks that are beneficial to the whole. This work offers a helpful approach for the implementation of the third level of inquiry, whereas earlier research on cognitive rehabilitation spaces concentrated on the first two levels of investigation.

The pandemic has increased the need for a healthy environment [72]; brought attention to the built environment's shortcomings and introspection [71]; and caused people to rethink how the built environment, industry norms, and policy standards should adapt to specific characteristics of the disease. The medical environment assessment system currently primarily consists of two aspects: health policy evaluation studies and health technology evaluation studies. The former uses effect evaluation as the evaluation standard after project implementation, while the latter uses rehabilitation efficacy as the evaluation standard after application. The findings of this study partially support the idea that the evaluation of the medical environment assessment system should take into account the habits, physiological requirements, and pathological traits of patients.

*5.3. Limitations*

The abundance and variety of the case studies are this paper's main drawback. The Guangzhou Yuexiu Elderly Service Center is a "day care" model, and as such, its environmental elements, such as the complexity of care, specifics of environmental design, and changes in patient behavior, may differ from those of a "full-day care" nursing home. The effective suggestions offered in this paper for similar studies are (1) to increase the volume and number of case studies and enrich the diversity of cases. For example, the impact of environmental factors on cognitive disorder patients in nursing homes with different care models can be compared, and weights can be set for various types of environmental factors, or the rehabilitation space in different models such as home care and community care can be compared. (2) Cognitive disorder patients can be studied by differentiating age, occupation, and gender, and the impact of different population characteristics on the study results can be compared.

**6. Conclusions**

The design of spatial layouts that are advantageous to individuals being treated for cognitive disorders are examined in this research. The research process demonstrates that: The behavioral (roaming) and pathological (cognitive impairment) characteristics of cognitive patients are strongly correlated with environmental factors; it is practical to assist the rehabilitation of cognitive patients through reasonable spatial layout design;

and it is effective to enhance the quality of rehabilitation of cognitive patients through spatial design.

The study came to the following conclusions: (1) Markers can be used to guide the roaming paths of patients with cognitive disorders; (2) Adjusting nurse station locations is a significant factor affecting the walking distance of nurses' shifts; (3) The essence of hospital layout iteration is the process of change from spatial behavior orientation to high quality information exchange; (4) The ideal spatial layout for patients with roaming behavior due to cognitive disorders is a central nursing station. A multi-core axis radiation arrangement with a central nursing station serving as the core, and various graded nurse stations simultaneously set up for coordinated management are the optimal spatial layout for patients who exhibit wandering behavior. The following patterns are anticipated to emerge in future study on the design of rehabilitation environments, despite the fact that the elements impacting the condition's rehabilitation are varied: the humanization of design from the standpoint of medical patients; the orientation of design for healthy living patterns; and the spatial adaptation of diseased traits.

Cognitive disorder cannot be cured [27], and experience from the past and the present has demonstrated that the best treatments for cognitive impairments at this time are early intervention and delay. Spatial interventions have the benefit of being relatively "low cost, effective, easy to apply, sustainable, and widely available," at least when compared to costly nurse training therapies and pharmaceutical treatments with side effects. Although there is now dispute on the efficacy of spatial interventions in the treatment of sickness, the particular context dictates the breadth and significance of our study, making the ongoing discussion about the notion of "ecological medicine" in sustainable design vital [73].

Due to the complexity of the rehabilitation mechanism of action [4], the limitation of this paper is that it concentrates on exploring the spatial layout factors that influence the rehabilitation of the condition, lacking more consideration of other factors, but also making the results of this study more relevant. The value of this paper is that it provides a reference path for the realization of environmental rehabilitation theory, complements the weaker part of previous studies (space and cognition), and elaborates the process of environmental modification, which provides a reference for the practice of designing rehabilitation environments for patients with cognitive disorders and similar populations. Because the methods and thought process in this study are based on applying lessons learned from prior successful experiences and cases, the conclusions are logically plausible. The impact of the outbreak has, however, restricted our team's many attempts to visit and seek partnership with pertinent medical institutes. The conclusions of this research need more real-world applications and time to further establish because the formation of creative theories and methodologies depends on the support and empirical support of a significant amount of case data.

The team's future research will focus on incorporating other variables that affect the recovery of cognitive disorders as part of the study, and further exploring whether improvements in the acoustic, light, and olfactory environments are also beneficial to cognitive recovery. The next research plan will be to seek to form cooperation with institutions such as nursing homes or retirement community service centers to work towards building a healthy habitat environment and to apply this theory to design practice.

**Author Contributions:** Conceptualization, W.L. and Z.D.; methodology, W.L. and Z.D.; software, W.L. and H.L.; validation, W.L., D.H.C.T. and Z.D.; formal analysis, W.L., Y.L. and H.L.; investigation, W.L. and Y.L.; resources, Y.L. and H.L.; data curation, Y.L. and H.L.; writing—original draft preparation, W.L.; writing—review and editing, W.L., Z.D., D.H.C.T. and K.W.Y.; visualization, W.L.; supervision, W.L., D.H.C.T., Z.D. and K.W.Y.; project administration, D.H.C.T., Z.D. and K.W.Y.; funding acquisition, W.L., D.H.C.T., Z.D. and K.W.Y. All authors have read and agreed to the published version of the manuscript.

**Funding:** This research was funded by the Key Project of Humanities and Social Sciences in General Universities of Guangdong Provincial Education Department (Grant No. 2019WZDXM007); The Youth Fund for Humanities and Social Sciences Research of the Ministry of Education (Grant No. 20YJC760135).

**Institutional Review Board Statement:** Not applicable.

**Informed Consent Statement:** Not applicable.

**Data Availability Statement:** Some or all data, models, or code generated or used during the study are available from the corresponding author by request.

**Acknowledgments:** The authors would like to thank the Guangdong Provincial Department of Education for their financial support and the staff of the Yuexiu Elderly Service Center in Guangzhou for their help. Authors would also like to express their gratitude to Lixiang Huang of Guangdong Technology College for her contribution to parts of this paper.

**Conflicts of Interest:** The authors declare no conflict of interest.

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
