# Peer review of "Study on the Design Strategy of Rehabilitation Space for Patients with Cognitive Disorders Based on the Environmental Adaptation of Disease Symptoms"

_sustainability, doi:10.3390/su141912590_

Round 1
Reviewer 1 Report
This article is interesting and the presentation very well organized. The main point of criticism is the link with the sustainability - in a broad sense all the improvements are in direction of sustainability, or, if an improvement exist, a sustainability of the intervention is demonstrated by this success. But it is not enough here, and we refer to environmental sustainability.
Specific remarks:
- please separate all the references with a space before, in the first half of your paper
-line 66 you have a comparison ... than ... but is not clear with what
- 71 - domestic scholar?
- 113 - explain the meaning os MCI - it alo included in the abstract - that is not easy to understand what it is, if you are not a specialist.
- 124 - briefly describe the technique
All the explanations of the changes done are well illustrated in chapter 3 - the assessment is only a theoretic calculation , assuming the positive effects - it is not quoted any practical experience or inquiry or experiments, even in your reference hospital.
The resulta are all theoretic based on assumptions - you should give an idea of what is happening in the reality
- line 370 - I created ?
- line 479 - 508 Here I think you should separate this text and include a chapter "limitations"
- line 511 Here is the most critical part, to be carefully examined
Here you say
"The research process demonstrates that: the behavioral (roaming) and pathological (cognitive impairment) characteristics of cognitive patients are strongly correlated with environmental factors" --> I don't see the demonstration of this
"it is practical to assist the rehabilitation of cognitive patients through reasonable spatial layout design" --> this is common sense, well, you designed adequate spaces for caregivers
"and it is effective to enhance the quality of rehabilitation of cognitive patients through spatial design" --> I would like to see a demonstration that there is an enhancement of the quality - based on previous researches you can surely suppose that this will happen - this means that your hypothesis is that it will enhance.
Your study came to a number of conclusions that according to your hypothesis will be positive, and will be demonstrated in future studies including ...... you should here offer an idea of the next steps to support you idea, including working with nurses, medical personnel and patient, if possible,
thank you for your work
Author Response
Point 1: The main point of criticism is the link with the sustainability - in a broad sense all the improvements are in direction of sustainability, or, if an improvement exist, a sustainability of the intervention is demonstrated by this success. But it is not enough here, and we refer to environmental sustainability.
Response 1: Thank you very much for pointing out the revisions to this article. In fact, we believe that this article is very much in line with the "Sustainability" journal, and that it can contribute to the construction of a better living environment, including rehabilitation landscape construction, urban road planning, medical architectural design, and healthy interior design. The impact of the epidemic has to some extent influenced the direction of our selection, and we hope to build a relationship between the environment and human health, and to illustrate the possibilities of the environment to assist patients in their recovery. Therefore, this topic falls under the category of healthy environment research. Although the direction of this selection is challenging, the authors involved in this paper are working toward the direction and goal of achieving ecological sustainability.
Point 2:please separate all the references with a space before, in the first half of your paper
Response 2: Thank you very much for pointing out your comments on citation formatting and detail adjustments, we have checked all citations in the text and have adjusted them accordingly.
Point 3:line 66 you have a comparison ... than ... but is not clear with what.
Response 3: Lines 67-70, thank you very much for your suggestions regarding the wording expression of the article. There may have been language expression issues that made it difficult for you to read, and we believe this may have happened to other readers as well, so we have adjusted it to read as follows: studies have shown that patients who heal in a properly designed natural environment do better in psychological adjustment and physical recovery than in an undesigned natural environment.
Point 4:- 71 -domestic scholar?
Response 4: line 74, thank you very much for the correction of the wording error in this article. In fact, we wanted to say "Chinese scholars" and we have corrected it.
Point 5:- 113 - explain the meaning os MCI - it alo included in the abstract - that is not easy to understand what it is, if you are not a specialist.
Response 5: Lines 18-19, thank you very much for pointing out the problem with the definition of terms in this paper. In fact, the definition of "MCI" is described in subsection 2.1 of this paper, but as you said, a description of it in the abstract would have given the reader a better understanding of the paper early on in the reading. We have already defined it in the abstract. The contents are as follows: Cognitive disorders cannot be cured, and the proposed MCI stage provides a window of opportunity for early intervention of the condition. MCI is a high-risk potential conversion state prior to a diagnosis of cognitive disorder, where the person still has the ability to live but with the presence of cognitive damage.
Point 6:- 124 - briefly describe the technique
Response 6: Sorry, this is a method and not a technique, we have replaced "technique" with "method" in line 129. The method is then described in detail in lines 149-155 and Table 1 of this paper. The contents are as follows: In order to achieve this, Hamilton developed the "environment-behavior-neuroscience" trinity sociological research method (Table 1). Its general steps are as follows: 1) thoroughly understand the subject's behavior using a variety of techniques, such as behavioral observation and expert consultation; 2) collect indicators of the patient's health variables using instruments and create a link between these indicators and behavioral variables.
Table 1. A trinity of environmental-behavioral-neuroscientific model of sociality research.
|
Environment-Behavior |
Neuroscience |
Design |
||
|
Variables, research methods and techniques in various fields |
||||
|
Behavioral outcomes |
Performance Results |
Neuroscience Factors |
Physiological factors |
Physical environment elements |
|
Observation method, photo documentation, self-reporting, etc. |
Clinical records, performances, expert evaluations, etc. |
PET scan, MRI, ERP evoked potentials, etc. |
Testing physiological responses, such as cortisol testing, blood pressure testing |
Describe environment specific Features, such as layout, scale, etc. |
Point 7:All the explanations of the changes done are well illustrated in chapter 3 - the assessment is only a theoretic calculation , assuming the positive effects - it is not quoted any practical experience or inquiry or experiments, even in your reference hospital.The result are all theoretic based on assumptions - you should give an idea of what is happening in the reality
Response 7: Thank you very much for your confusion regarding the argumentative part of this paper, which was designed to reason and assess based on the pathological characteristics (behavioral characteristics and cognitive impairment) of the study population.
- To this end, we conducted a field survey of the Guangzhou Yuexiu Elderly Service Center, interviewed the elderly and caregivers, etc., and synthesized their oral accounts, caregiving experiences, field situations, literature, and case studies to jointly guide the study of this paper.
- In addition, the reasoning process of this study is somewhat experimental in nature, comparing among multiple scenarios, perspectives and levels so as to screen the optimal strategy, which is a method to verify the effectiveness of environmental modification through a continuous improvement and optimization, and is evaluated by comparing the change values of environmental variables.
Point 8:- line 370 - I created ?
Response 8: Lines 381-382. Thank you very much for pointing out the problem with the use of wording in this paper, which we have corrected. In fact, what we wanted to convey is that this floor is divided into an 11x20 gravitational network.
Point 9:- line 479 - 508 Here I think you should separate this text and include a chapter "limitations"
Response 9: Line 512, thank you very much for your comments on the restructuring of this article. Yes, I agree with you on the content of the subsection read here. The restructuring has improved the readability of the article.
Point 10:- line 511 Here is the most critical part, to be carefully examined.Here you say
- "The research process demonstrates that: the behavioral (roaming) and pathological (cognitive impairment) characteristics of cognitive patients are strongly correlated with environmental factors" --> I don't see the demonstration of this
- "it is practical to assist the rehabilitation of cognitive patients through reasonable spatial layout design" --> this is common sense, well, you designed adequate spaces for caregivers
- "and it is effective to enhance the quality of rehabilitation of cognitive patients through spatial design" --> I would like to see a demonstration that there is an enhancement of the quality - based on previous researches you can surely suppose that this will happen - this means that your hypothesis is that it will enhance.
Response 10: Lines 195-216, thank you very much for your review and attention to the conclusion section of this paper. Indeed, this paper illustrates the correlation between roaming, cognition, and environment in subsection 3.1.1. To strengthen this this logical relationship, we have added Table 3 in line 216 to illustrate it. Besides, some of these examples (by installing strolling corridors and enriching the stimulating environment, thereby reducing roaming behavior, increasing environmental engagement, and slowing cognitive decline) provide partial support for the idea that spatial design can improve the quality of recovery for people with cognitive disorders.
Point 11:Your study came to a number of conclusions that according to your hypothesis will be positive, and will be demonstrated in future studies including ...... you should here offer an idea of the next steps to support you idea, including working with nurses, medical personnel and patient, if possible,
Response 11: Lines 559-575, your suggestions for further in-depth research in the future of this paper are greatly appreciated and we will add to this paper accordingly.
The additions are as follows:
Due to the complexity of the rehabilitation mechanism of action [4], the limitation of this paper is that it concentrates on exploring the spatial layout factors that influence the rehabilitation of the condition, lacking more consideration of other factors, but also making the results of this study more relevant. The value of this paper is that it provides a reference path for the realization of environmental rehabilitation theory, complements the weaker part of previous studies (space and cognition), and elaborates the process of environmental modification, which provides a reference for the practice of designing rehabilitation environments for patients with cognitive disorders and similar populations. Due to the epidemic, several applications for visits and opportunities to seek collaboration with relevant medical institutions have been limited.
The team's future research will focus on incorporating other variables that affect the recovery of cognitive disorders as part of the study, and further exploring whether improvements in the acoustic, light, and olfactory environments are also beneficial to cognitive recovery. The next research plan will be to seek to form cooperation with institutions such as nursing homes or retirement community service centers to work towards building a healthy habitat environment and to apply this theory to design practice.
All authors of this paper would like to express their sincere gratitude to you for your corrections, as each of your suggestions is intended to improve the quality of this research paper. We would also like to thank the editor and staff responsible for this paper for their review work, which made it an honor to have this study published in the journal Sustainability.
Reviewer 2 Report
Dear authors,
Congratulations on the completion of this research work, for your time and dedication.
My comments are very positive about your research.
I congratulate you on the conceptualisation of the problem, the design and method, as well as the discussion of the debate and conclusion. Very elaborate.
I also congratulate you on the in-depth discussion and broad conceptualisation, as well as the detailed method followed.
I will now make some suggestions for improvement, with the aim of improving your citations, downloads, visits etc.
-I suggest that you incorporate as much as you can at the end of the discussion:
a) what are the theoretical implications of this work for the scientists who read this work, for the theoreticians in the field or colleagues.
b) what practical implications does this work have for older people?
c) strengths of their work in relation to other studies.
d) future lines of research arising from this work that need to be pursued.
I hope that in this way you will get a better visibility! congratulations!
My sincere congratulations for the work.
Author Response
Point 1: I will now make some suggestions for improvement, with the aim of improving your citations, downloads, visits etc. I suggest that you incorporate as much as you can at the end of the discussion:
- a) what are the theoretical implications of this work for the scientists who read this work, for the theoreticians in the field or colleagues.
- b) what practical implications does this work have for older people?
- c) strengths of their work in relation to other studies.
- d) future lines of research arising from this work that need to be pursued.
I hope that in this way you will get a better visibility! congratulations!
My sincere congratulations for the work.
Response 1: Lines 559-575, your suggestions for further in-depth research in the future of this paper are greatly appreciated and we will add to this paper accordingly.
The additions are as follows:
Due to the complexity of the rehabilitation mechanism of action [4], the limitation of this paper is that it concentrates on exploring the spatial layout factors that influence the rehabilitation of the condition, lacking more consideration of other factors, but also making the results of this study more relevant. The value of this paper is that it provides a reference path for the realization of environmental rehabilitation theory, complements the weaker part of previous studies (space and cognition), and elaborates the process of environmental modification, which provides a reference for the practice of designing rehabilitation environments for patients with cognitive disorders and similar populations. Due to the epidemic, several applications for visits and opportunities to seek collaboration with relevant medical institutions have been limited.
The team's future research will focus on incorporating other variables that affect the recovery of cognitive disorders as part of the study, and further exploring whether improvements in the acoustic, light, and olfactory environments are also beneficial to cognitive recovery. The next research plan will be to seek to form cooperation with institutions such as nursing homes or retirement community service centers to work towards building a healthy habitat environment and to apply this theory to design practice.
All authors of this paper would like to express their sincere gratitude to you for your corrections, as each of your suggestions is intended to improve the quality of this research paper. We would also like to thank the editor and staff responsible for this paper for their review work, which made it an honor to have this study published in the journal Sustainability.

Round 2
Reviewer 1 Report
I reviewed the revised version and I find it positive, having accepted all the suggested changes.
Author Response
Point 1: I reviewed the revised version and I find it positive, having accepted all the suggested changes.
Response 1: Many thanks to the reviewers for pointing out the details of the revisions to this article and for recognizing the quality of the article. In conjunction with the comments from the academic editor, the following additions and clarifications have been made in the conclusion section of the article regarding the feasibility of the study results.
Lines 571-577, The value of this paper is that it provides a reference path for the realization of environmental rehabilitation theory, complements the weaker part of previous studies (space and cognition), and elaborates the process of environmental modification, which provides a reference for the practice of designing rehabilitation environments for patients with cognitive disorders and similar populations. Because the methods and thought process in this study are based on applying lessons learned from prior successful experiences and cases, the conclusions are logically plausible. The impact of the outbreak has, however, restricted our team's many attempts to visit and seek partnership with pertinent medical institutes. The conclusions of this research need more real-world applications and time to further establish because the formation of creative theories and methodologies depends on the support and empirical support of a significant amount of case data.
The authors have checked the citation of the article references several times.
All authors of this paper would like to express their sincere gratitude to you for your corrections, as each of your suggestions is intended to improve the quality of this research paper. We would also like to thank the editor and staff responsible for this paper for their review work, which made it an honor to have this study published in the journal Sustainability.